# The Effect of Dietary Supplementation with Zinc Amino Acids on Immunity, Antioxidant Capacity, and Gut Microbiota Composition in Calves

**DOI:** 10.3390/ani13091570

**Published:** 2023-05-08

**Authors:** Pengxia Hou, Bo Li, Yan Wang, Dan Li, Xiaoyu Huang, Wenyang Sun, Xiaojun Liang, Enping Zhang

**Affiliations:** 1College of Animal Science and Technology, Northwest A&F University, Xianyang 712100, China; houpenxia@163.com (P.H.);; 2Ningxia Academy of Agriculture and Forestry Science, Institute of Animal, Yinchuan 750002, China

**Keywords:** calves, zinc amino acids, gut microbiota, *Bacteroides*, *Prevotellaceae-UCG-003*

## Abstract

**Simple Summary:**

The stable microecological system in animal intestines is beneficial for the healthy growth of calves, and as zinc is an important trace element in the body, the question of how to better promote its absorption by the body has become an important research hotspot. Research has shown that an appropriate amount of zinc addition can promote the growth and proliferation of beneficial bacteria in the body’s intestines and inhibit the proliferation of harmful bacteria. This study showed that adding 80 mg/kg zinc amino acids to the diet can improve the growth performance, immune performance, antioxidant capacity, intestinal microbiota enrichment, and the intestinal microbial environment, as well as promote the healthy growth, of calves.

**Abstract:**

The aim of this study was to investigate the effect of dietary supplementation with zinc (Zn) amino acids at different concentrations on immunity, antioxidant capacity, and gut microbiota composition in calves. Twenty-four one-month-old healthy Angus calves of comparable body weight were randomly divided into three groups (four males and four females in each group) based on the amount of Zn supplementation added to the feed the animals received: group A, 40 mg/kg DM; group B, 80 mg/kg DM; and group C, 120 mg/kg DM. The experiment ended when calves reached three months of age (weaning period). The increase in dietary Zn amino acid content promoted the growth of calves, and the average daily weight gain increased by 36.58% (*p* < 0.05) in group C compared with group A. With the increase in the content of dietary Zn amino acids, the indexes of serum immune functions initially increased and then decreased; in particular, the content of immunoglobulin M in group A and group B was higher than that in group C (*p* < 0.05), whereas the content of interleukin-2 in group B was higher than that in the other two groups (*p* < 0.05). In addition, the content of superoxide dismutase and total antioxidant capacity in the serum of calves in group B was higher than that in group C (*p* < 0.05), and the MDA level was lower than in group C (*p* < 0.05). Moreover, alpha diversity in the gut microbiota of calves in group B was higher than that in group A and group C (*p* < 0.05); the dominant phyla were *Firmicutes* and *Bacteroidota*, whereas the dominant genera were *Unclassified-Lachnospiraceae* and *Ruminococcus*. Linear discriminant analysis showed that the relative abundance of *Bacteroides* in the gut microbiota of calves in group B was higher than that in group A, and the relative abundance of *Prevotellaceae-UCG-003* was higher compared to that in experimental group C. Thus, dietary supplementation of 80 mg/kg of Zn amino acids to calves could improve the immune function and antioxidant capacity, as well as enrich and regulate the equilibrium of gut microbiota, thus promoting the healthy growth of calves.

## 1. Introduction

The alternance between dynamism and stability in the gut microbial ecosystem impacts the animal host, and the type of microorganisms in the gut is closely associated with the host’s metabolism, immunity, as well as tissue and intracellular homeostasis [1]. A stable gut microbiota can regulate animal health by promoting nutrient digestion and absorption, maintaining intestinal barrier integrity, and enhancing organism immune function [2]. On the contrary, disruptions to the gut microbiota can cause metabolic disorders, such as diabetes and non-alcoholic fatty liver, which hinders animal growth and development [3,4]. Therefore, maintaining a stable gut microbiota, being beneficial to animal growth and development, is one of the focuses of animal husbandry practices.

Zinc (Zn) is a component of more than forty different enzymes and two-hundred enzyme activators, and thus plays a key role in animal growth and development [5]. Inorganic zinc (IOZ) is the main form of Zn added to feed. However, due to its low absorption rate, unstable chemical properties, and environmental pollution capacity, IOZ has been gradually replaced by organic zinc (OZ), which has good palatability, high chemical stability, lipophilicity, and bioavailability [6,7]. Previous studies have shown that the addition of OZ to feed (protein zinc, amino acid chelated/complexed zinc, and polysaccharide zinc [8,9]) can alleviate diarrhea [10], improve immune function [11] and antioxidant capacity [12], promote intestinal health [13], and enhance reproductive performance [14], thus supporting the healthy growth of animals.

Different sources and doses of Zn affect the gut microbiota of animals. The appropriate amount of Zn can promote the growth and proliferation of beneficial bacteria while inhibiting harmful bacteria in the gut. Yu et al. found that the addition of 120 mg/kg of OZ to the diet increased the relative abundance of Firmicutes, while decreasing the relative abundance of Proteobacteria in the intestinal tract of fattening pigs [15]. Xie et al. found that the addition of 100 mg/kg of zinc chitosan to the diet of weaned piglets increased the number of *Lactobacillus* in the cecum and colon, while decreasing the number of *Escherichia coli* and *Salmonella* [16]. Broom et al. and Højberg et al. also showed that a zinc-rich diet decreased the number of anaerobic bacteria and *Lactobacillus* in the ileum of piglets, but increased the number of *Escherichia coli* [17,18]. Few studies have been conducted on the application of OZ on calves, especially considering its effects on gut microorganisms.

Therefore, the aim of this study was to analyze the effects of supplementation with Zn amino acid complexes at different concentrations on the immunity, antioxidant capacity, and gut microbiota composition of calves. High-throughput second-generation 16S rRNA sequencing was used for assessing bacterial community composition in the guts of calves. The findings discussed herein provide a theoretical basis for investigating the appropriate level of Zn amino acid complex dietary supplementation to calves.

## 2. Materials and Methods

### 2.1. Animals and Experimental Design

All animal procedures used in this study were previously approved by the Animal Care and Use Committee (IACUC) of the Institute of the Ningxia Academy of Agriculture and Forestry Sciences, China (DK2019060). The experiment was conducted at the institutional experimental farm.

Twenty-four Angus calves (30 ± 5.10 days of age; 69.34 ± 7.67 kg in body weight) were included in the study. Calves were housed from birth with their mothers in pens equipped with a separate feeding trough, which enabled the obtainment of supplementary pellets from ten days of age. Animals were equally divided into three experimental groups based on the level of Zn supplementation (eight animals per group): (i) group A, 40 mg/kg DM; (ii) group B, 80 mg/kg DM; and (iii) group C, 120 mg/kg DM. Zn was added in the form of Zn amino acid complexes (Zn content, 120 g/kg; total amino acid content, ≥21%), replacing the zinc in the basic feed premix (zinc content, 80 mg/kg; added in the form of zinc sulfate) and mixed with other raw materials to make pellet feed. The entire experiment lasted for 60 days.

The animals were numbered and fed with pellets containing different amounts of Zn every day. The cow pen was equipped with a calf feeding pen, allowing only calves to freely enter and exit for feeding. Feeding of the calves with pellets occurred twice daily, at 9:00 and 16:00. Pens were equipped with a sink, cleaned and disinfected weekly, and the calves had access to hay and water ad libitum. The feed dietary composition and nutrient levels of the basal diet are shown in Table 1.

### 2.2. Measurements of Growth Performance and Immunity Indexes

#### 2.2.1. Growth Performance

Calves were weighed on an empty stomach in the morning at initial body weight (one month old) and final body weight (three months old), and the average daily gain (ADG) was calculated. Calf pellet input and the remaining amount were weighed for three consecutive days weekly to determine calf feed intake. The average daily dry matter intake and feed-to-weight ratio were also calculated.

#### 2.2.2. Serum Indicators of Immune and Antioxidant Performance

At 7 a.m. the next day after the experiment, 5 mL of blood was collected from the calf tail vein using a vacuum anticoagulant device, left to stand, centrifuged at 3500× *g* for 5 min, and divided into two samples in 1 mL centrifuge tubes, followed by storage at −20 °C in 1.5 mL Eppendorf tubes until further analysis. 

Serum concentrations of immunoglobulin A (IgA), immunoglobulin G (IgG), immunoglobulin M (IgM), and interleukin-2 (IL-2) were determined using an Enzyme-Linked Immunosorbent Assay in a Multiskan FC Microplate reader (Thermo Fisher Scientific, Waltham, MA, USA). 

Colorimetric determination of serum glutathione peroxidase (GSH-Px), superoxide dismutase (SOD), total antioxidant capacity (T-AOC), and malondialdehyde (MDA) was conducted using an ELISA analyzer Rayto RT-6100 (Shenzhen Raydu Life Science Co., Ltd., Shenzhen, China).

### 2.3. Metagenomic Analysis of the Gut Microbiome

#### 2.3.1. DNA Extraction and 16S rRNA Gene Sequencing

At the end of the experiment, fecal samples of the calves supplemented with Zn amino acid complexes at different concentrations were collected by rectal extraction. Fecal samples were placed in 2 mL lyophilization tubes, which were immediately placed into liquid nitrogen and then stored at −80 °C until further analysis. Total DNA extraction and PCR amplifications were performed by QingKe Bio, Ltd. (Beijing, China). 

#### 2.3.2. Sequencing Data Processing

The quality control of raw sequencing reads was conducted using Trimmomatic software v.0.33. Primer sequences were identified and removed using cutadapt software 1.9.1 to obtain clean sequencing reads without adapters. Clean reads were then spliced by overlapping using Usearch software v.10. Spliced sequencing data were length-filtered according to the length ranges of different genomic regions, and chimeric sequences were identified and removed to obtain the final effective sequencing reads using UCHIME software v.4.2. Reads were clustered at a 97.0% similarity level using Usearch software to obtain operational taxonomic units (OTUs) [20]. Classification of obtained OTUs was conducted using QIIME2 software. Alpha-diversity indexes were calculated [21], and beta-diversity analysis was performed using QIIME software. The taxonomic annotation of species in each feature was performed using a plain Bayesian classifier with the SILVA database. The community composition of each sample was determined at each taxonomic level (phylum, class, order, family, genus, and species), and species at different taxonomic levels were determined using QIIME software. Abundance tables were generated for different taxonomic levels using QIIME software, while the microbial community composition of samples at each taxonomic level was constructed using R language. Finally, microbial biomarkers were identified in RStudio based on linear effect size (Lefse) analysis using the MicroBiotaProcess program package [22].

### 2.4. Data Analysis

Data were expressed as means ± standard deviations. Body weights, immune function indexes, and other data were initially processed in Microsoft Excel software. One-way ANOVA was performed in SPSS software (version 22.0) (IBM Corp., Armonk, NY, USA), and Duncan’s test was used for multiple comparisons, in which *p* < 0.05 indicated significant differences. The significance of the alpha-diversity of gut microbiota composition in different experimental groups was determined using *t*-test in QIIME2 software (version 2.0). Principal coordinate analysis (PCoA) was used for analyzing beta-diversity measurements. Linear discriminant analysis (LDA) effect size (LEfSe) analysis was performed using the MicroBiotaProcess program package in RStudio to determine microbial biomarkers, including the abundance of bacterial taxa from phylum to species level [21]. The thresholds for the false-discovery rate (FDR) and the log LDA score were 0.01 and 4, respectively.

## 3. Results

### 3.1. Growth Performance

As shown in Table 2, the final weight of calves showed an increasing trend with the increase in the concentration of Zn amino acid complex supplementation; the final weight of calves in group C was higher than in group A (*p* < 0.05); moreover, the average daily weight gain was 36.58% higher in group C compared to group A (*p* < 0.05).

### 3.2. Immune Function

As shown in Table 3, IgM content in group A and group B was higher than in group C (*p* < 0.05). Moreover, IL-2 content in group B was higher than in the other two groups (*p* < 0.05). 

### 3.3. Antioxidant Capacity

There was no significant difference in serum GSH-Px content among treatments, SOD and T-AOC levels were higher in group B compared to group C (*p* < 0.05), and the MDA level was lower in group A and group B compared to group C (*p* < 0.05) (Table 4).

### 3.4. Gut Microbiota Composition

#### 3.4.1. Sequencing Data Quality Control

As shown in Table 5, 1,200,040 reads were obtained in total, and 1,159,596 valid reads were obtained after quality control and the splicing of paired-end reads was conducted to remove low-quality reads and chimeras. The ratio of valid reads in all experimental groups of samples was more than 94%, and Q20 and Q30 values were above 99% and 96%, respectively, which indicated that sequencing results could adequately reflect the microbiota in the guts of calves receiving Zn supplementation.

#### 3.4.2. Number of OTUs 

As shown in Figure 1, 818 OTUs were identified in the three sample groups, and the OTUs uniquely assigned to group A, group B, and group C were 13, 3, and 5, respectively. OTUs shared by samples in the three groups corresponded to 89.40% of the total OTUs; in contrast, OTUs uniquely found in each sample group accounted for only 2.30% of the total OTUs. OTU compositions in fecal samples showed high similarity, indicating that the gut microbiota structure was relatively stable among calves receiving Zn supplementation at different concentrations.

#### 3.4.3. Sequencing Depth

As shown in Figure 2, the dilution curves of each sample group stabilized as sequencing depth increased, indicating that sequencing depth covered the bacterial species in the samples and that sequencing volume and depth were reasonable, such that they could be used for the analysis of bacterial population diversity.

#### 3.4.4. Alpha-Diversity Analysis

Table 6 shows that both group A and group B outperformed group C in terms of Simpson and Shannon indexes (*p* < 0.05).

#### 3.4.5. Beta-Diversity Analysis

PCoA results are shown in Figure 3A. Principal component 1 (PC1) contributed 28.78% to explanation of the variance of the samples, whereas PC2 contributed 21.25% to explanation of the variance of the samples. In addition, the samples between the three groups were separated, indicating large differences in the composition and structure of the gut microbiota of calves. In contrast, samples from the same groups clustered together, indicating small differences in intra-group diversity. 

Analysis of similarities (ANOSIM) results are shown in Figure 3B. The ANOSIM value between sample groups was R = 0.351 > 0, indicating that inter-group differences were greater than intra-group differences (*p* < 0.05). Collectively, the results indicated a high degree of reliability and reasonable grouping.

#### 3.4.6. Relative Abundance and Structure of Gut Microbiota at Phylum and Genus Levels

As shown in Figure 4 and Table 7, the dominant microbial phyla in each sample group were *Firmicutes*, *Bacteroidota*, and *Proteobacteria*; the dominant genera were *unclassified-Lachnospiraceae, Alloprevotella*, and *Ruminococcaceae-UCG-005*. Differences in species abundance between groups were analyzed using non-parametric statistics at different taxonomic levels, and the results are shown in Table 8. The relative abundance of *Lactobacillus* and *Faecalibaculum* was higher in group B compared to group A at the genus level (*p* < 0.05); the relative abundance of *Prevotellaceae-UCG-003*, *Faecalibaculum*, and *Bifidobacterium* was higher compared to group C (*p* < 0.05); and the relative abundance of *Bacteroides* was higher in group C than in group A (*p* < 0.05).

#### 3.4.7. LEfSe Analysis

Herein, the LEfSe score threshold was set to 4, and species with LDA scores ≥ 4 were considered the dominant species with very significant high abundance. 

As shown in Figure 5, species with significant differences in group B were Spirochaetacene, Bacteroidaceae, unclassified Prevotellaceae, Prevotellaceae-NK3B31-group, Treponema, Bacteroides, and Prevotellaceae-UCG-003. The relative abundance of Bacteroidaceae increased (*p* < 0.05) and the relative abundance of *Spirochaetaceae* and Treponema decreased (*p* < 0.05), whereas the relative abundance of Prevotellaceae-UCG-003 increased compared with group C (*p* < 0.05).

## 4. Discussion

Zinc is a crucial component for animal growth and development, being also crucial for preserving proper metabolism [23]. Recent studies have demonstrated the value of dietary supplementation with adequate levels of zinc to support animal development performance [24]. Pei et al. demonstrated that weaned pigs fed with 450 mg/kg of ZnO nanoparticles had increased growth performance [25]. In addition, Graget et al. reported that adding 20 mg/kg of ZnO and Zn methionine to feed promoted the growth of lambs, the Zn-meth group having considerably greater daily weight gain than the ZnO group [26]. Chang et al. showed that neonatal calves receiving 80 mg of Zn methionine daily had increased daily weight gain after 14 days of supplementation [27]. Moreover, the average daily weight gain of bull calves was greatly increased by daily supplementation with 0.45 g of Zn methionine complexes (80 mg of Zn) [28]. Thus, the results of the current study are consistent with those of previous studies, in that the average daily weight gain of calves increased as the level of dietary zinc amino acid supplementation increased; in particular, group C outperformed group A in terms of weight gain. Subsequently, we investigated the effect of Zn amino acids at different levels on the immune function and antioxidant capacity of calves to better determine the impact of Zn supplementation on calves’ health.

Animal health and metabolic state are reflected in blood markers. The levels of IgA, IgM, and IgG are significant markers of immune response, and the trace element Zn is intimately involved in the immune response [29,30,31,32,33]. The levels of IgM and IgG in the blood of calves increased after dietary supplementation with 80 mg/day of ZnO [27]. Wei et al. described that the levels of IgA, IgM, and IgG in the blood of calves showed an upward tendency with the increase in ZnO concentration (40–120 mg/kg) in the diet [34]. In the present study, no discernible effect was observed as a result of dietary supplementation with Zn amino acids on the levels of IgA and IgG in blood; in contrast, the level of IgM in group B was higher than that in groups A and C, which was inconsistent with the results of the above studies. This could be likely due to different sources of Zn used in this study, which led to differences in actual concentrations of Zn supplemented. IL-2 is a T-lymphocyte growth factor that plays an important role in the immune response [35]. Chen et al. revealed that feeding dairy cows with 60 mg/kg of Zn methionine dramatically raised the levels of IL-2 in the blood [36]. In the present study, the levels of IL-2 in the blood of calves increased initially and then decreased as the amount of Zn amino acids in the feed increased, thus suggesting that increasing dietary supplementation with 80 mg/kg DM of Zn amino acids will benefit the immune system of calves.

Antioxidative compounds are produced by the body in response to harmful external stimuli to withstand challenges [37]. The body’s potential for antioxidant defense is indicated by levels of T-AOC, which is composed of numerous antioxidative compounds and enzymes. SOD can neutralize a great number of free radicals generated in the body as a result of stressors and scavenge reactive oxygen species (ROS) [38]. Moreover, the level of cell damage is indicated by MDA concentration, which is a byproduct of lipid peroxidation in cell membranes. Several studies have demonstrated that Zn can increase the body’s capacity for antioxidant response, which is directly linked to the equilibrium of the body’s redox system [39,40]. Yu et al. revealed that serum T-AOC levels during the fattening stage were considerably greater in pigs receiving 90 mg/kg, 120 mg/kg, and 150 mg/kg of cysteamine chelated Zn compared to pigs receiving 60 mg/kg [15]. The addition of protein Zn to the diet at concentrations greater than 80 mg/kg dramatically raised serum SOD levels, as demonstrated by Kannan et al. [41]. In the present study, the levels of T-AOC and SOD in the blood of calves tended to increase and then decrease as the concentration of dietary Zn amino acids increased, whereas the serum MDA levels increased. Thus, these findings suggest that, under the conditions employed in the current experiment, moderate addition of Zn amino acids to the diet improved the antioxidant capacity of calves, while an excess of Zn decreased antioxidant capacity. Subsequently, we carried out correlation analysis of the gut microbiome of calves following the addition of Zn amino acids at different concentrations to further investigate the ideal amount of Zn amino acids to be supplemented in calf diets.

In order for calves to digest and absorb nutrients, the gut microbiota is essential, as it directly impacts calves’ health. When the gastrointestinal microbiota is unbalanced and prone to invasion by pathogenic bacteria, diarrhea and other intestinal disorders can occur [42]. Herein, the alpha-diversity of the gut microbiota of calves in group B was substantially greater than in groups A and C, showing that the addition of 80 mg/kg DM of Zn amino acids to the diet improved the variety of the gut microbial composition. Jensen et al. showed that the variety of the rectal microbiota was not affected by the addition of 2500 mg/kg of ZnO to the diet of weaned pigs [43] which was not in accordance with the results of the present study, possibly due to the differences in feed composition, nutrition level, feeding environment, and Zn source.

*Firmicutes* play a significant role in how well proteins and carbohydrates are absorbed [44]. In contrast, *Bacteroides* primarily target non-fibrous materials for digestion, but species within this phylum can also colonize the gut to reduce the adherence of invasive pathogens [45,46]. It is known that obesity level is reflected in the ratio of *Firmicutes* and *Bacteroides* in the gut microbiota [47]. Tian et al. found that the dominant phyla in the gut microbiota of calves were *Firmicutes* and *Bacteroidetes* [48]. In the present study, the dominant microbial phyla and genera in the fecal samples of calves receiving Zn supplementation were not impacted by dietary levels of Zn amino acids. In addition, *Ruminococcus* and *Bacteroides* were the dominant genera in the gut microbiota of calves in all sample groups. The phylum *Bacteroides* is primarily involved in the hydrolysis of proteins and the degradation of carbohydrates [46]. According to Chang et al., the relative abundance of *Bacteroides* in the gut microbiota of calves increased considerably after receiving daily supplementation with 457 mg (80 mg/day) of Zn methionine [27]. In the present study, the relative abundance of Bacteroides in calves of group C was considerably higher than in group A. Moreover, supplementation with 120 mg/kg DM of Zn amino acids increased the utilization of protein and carbohydrate in the feed, thus promoting the growth of calves, based on average daily weight gain data.

The immune system and *Lactobacillus* are closely associated [49]. The production of lactic acid prevents the growth and multiplication of pathogens such as *Colidextribacter*, and it has been demonstrated that *Lactobacillus* can lower local pH in the gastrointestinal tract [50]. To maintain the environmental acid and prevent the growth of infective microbial species in the gut, *Lactobacillus* can also convert lactic acid to butyrate. Hou et al. found that feeding weaned piglets with chitosan-chelated zinc (100 mg/kg zinc) increased the relative abundance of *Lactobacillus* in the cecum [13]. In addition, *Faecalibaculum* is a probiotic species that produces a natural antibiotic that can improve health in the colon by generating metabolites, such as butyric acid and short-chain fatty acids, among others [51]. In the present study, the relative abundances of *Lactobacillus* and *Faecalibaculum* were considerably higher in group B than in group A, suggesting that feeding calves with 80 mg/kg of Zn amino acids was more effective in improving intestinal health.

*Prevotella* degrade starch into monosaccharides and other non-cellulose polysaccharides to provide energy to the body [52] as well as to boost nutritional digestion and absorption [53,54,55]. However, this degradation might promote inflammation [56]. In addition, *Bifidobacterium* is another physiologically significant intestinal bacterium that can interact with immune cells and control the immune system [57,58]. In another study, compared to the group receiving no dietary addition of Zn methionine, it was revealed that the addition of 70 mg/kg and 140 mg/kg of Zn methionine considerably increased the abundance of *Bifidobacterium* in the cecum of laying hens [59]. In the present study, a considerably higher relative abundance of *Prevotella-UCG-003* and *Bifidobacterium* was found in group B compared to group C. Thus, the addition of Zn amino acids to the feed boosted the immune system of calves, while it also promoted nutrient absorption. Therefore, it was established that the optimal amount of Zn amino acids to be added to the feed given to calves was 80 mg/kg DM.

Furthermore, using LEfSe analysis, significant differences in the abundance of *Spirochaetacene*, *Treponema, Bacteroides, Prevotellaceae-UCG-003,* and other species were found in group B. Among these, *Spirochaetacene* and *Treponema* are Gram-positive intestinal pathogenic bacteria [60,61]. By interacting with the immune system, *Bacteroides* stimulate T-cell-mediated responses and prevent potentially harmful bacteria from colonizing the gut [62,63]. Chang et al. demonstrated that 104 mg/day of ZnO supplementation decreased the incidence of diarrhea in calves and raised the relative abundance of *Bacteroides* in seven-day-old calves [27]. In the present study, compared to group A, a higher relative abundance of *Bacteroides* and a lower relative abundance of *Spirochaetaceae* and *Treponema* were found in group B, which suggests that including Zn amino acids in the diet of calves will improve the body’s immune system.

Finally, *Prevotellaceae-UCG-003* was first described by Koh et al. as having the capacity to control intestinal inflammation by triggering dendritic cells through succinate synthesis [64]. In the present study, *Prevotellaceae-UCG-003* was found in greater relative abundance in group B than in group C, showing that dietary addition of the Zn amino acids at the levels used in group B was more effective in enhancing host immune function. Collectively, it can be stated that the addition of 80 mg/kg DM of Zn amino acids to feed improved intestinal health and led to an increase in the relative abundance of beneficial microorganisms in the guts of calves.

## 5. Conclusions

In the present study, the levels of IgM, IL-2, T-AOC, and SOD in the blood of calves in Experiment B were considerably greater than those in Experiments A and C, as well as the variety of fecal microbiota; Group B’s average daily weight gain rose by 13.41% compared to Group A.Additionally, LEfSe scores revealed a much higher relative abundance of *Prevotellaceae-UCG-003* in group B than in group C. Moreover, the relative abundance of *Bacteroides* in group B was higher than in group A. Thus, the findings discussed herein highlighted that the addition of 80 mg/kg DM of Zn amino acids to the diet of calves could improve immune response and antioxidant capacity, enrich the gut microbiota, regulate the equilibrium of the gut microbiota, and promote the healthy growth of calves.

## Figures and Tables

**Figure 1 animals-13-01570-f001:**
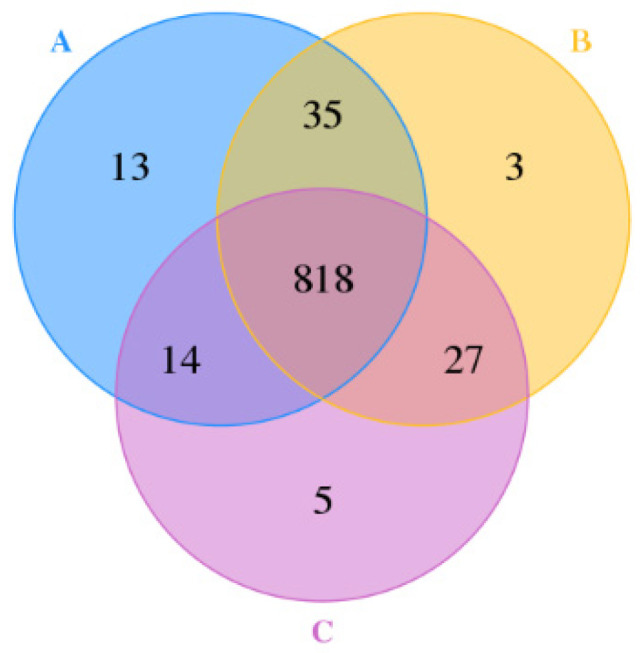
Venn diagram of shared and unique operational taxonomic units (OTUs) in the bacterial community in the guts of calves receiving zinc amino acid complexes as dietary supplementation. A, B, and C refer to group A, group B, and group C, respectively.

**Figure 2 animals-13-01570-f002:**
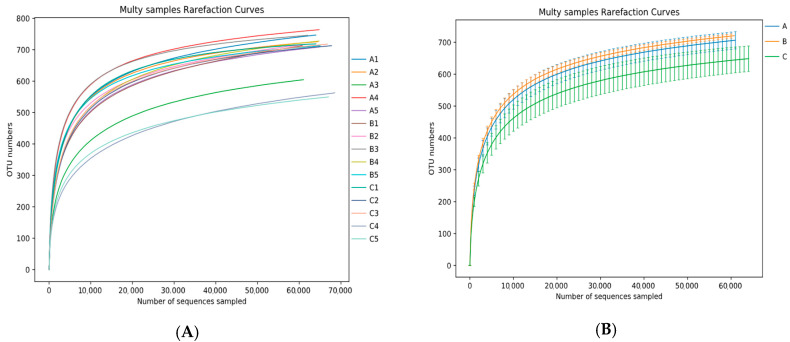
Single-sample dilution curves (**A**) and group dilution curves (**B**). Horizontal coordinates indicate the number of randomly selected sequencing strips, whereas vertical coordinates indicate the number of features obtained based on the number of sequencing strips. Each curve in (**A**) refers to a single sample; each curve in (**B**) refers to a sample group.

**Figure 3 animals-13-01570-f003:**
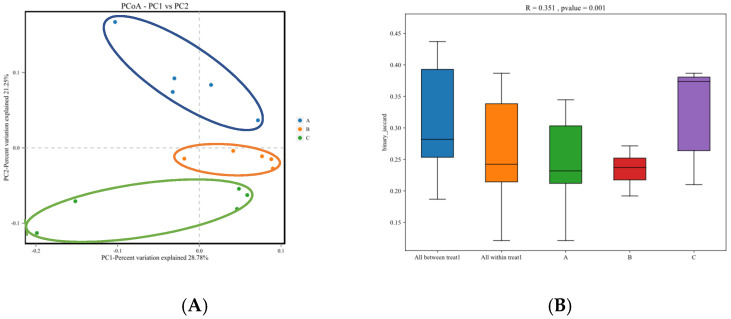
Principal coordinate analysis (PCA) plots (**A**) and analysis of similarities (ANOSIM) chart (**B**) of the gut microbiome in fecal samples of calves receiving zinc supplementation at different levels. Note: A, B, and C in the figure represents Group A, Group B, and Group C respectively.

**Figure 4 animals-13-01570-f004:**
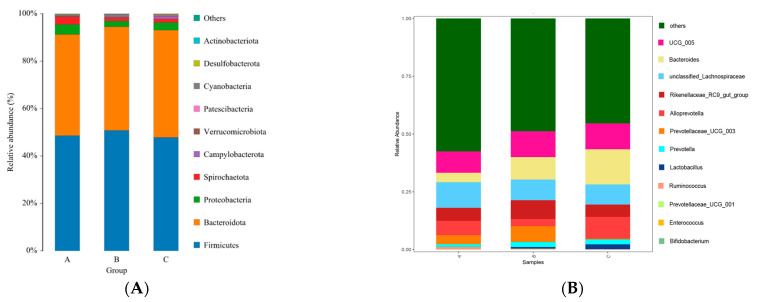
Relative abundance of gut microbiota at the phylum level (**A**) and at the genus level (**B**) in calves receiving zinc supplementation at different levels. Note: A, B, and C in the figure represents Group A, Group B, and Group C respectively.

**Figure 5 animals-13-01570-f005:**
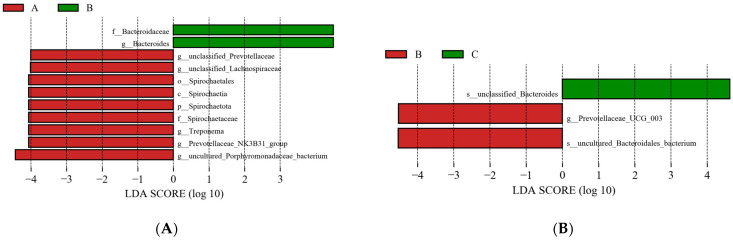
(**A**,**B**) linear discriminant analysis (LDA) effect size (LEfSe) analysis of gut microbiota of calves receiving zinc supplementation at different levels. Note: A, B, and C in the figure represents Group A, Group B, and Group C respectively.

**Table 1 animals-13-01570-t001:** Composition and nutrient levels of the basal diet offered to calves included in the experiment (dry matter basis %).

Ingredient	Content	Nutrient Levels	Content
Flaked Corn	58.60	Combined net energy ^2^ (MJ/kg)	5.69
Soybean meal (43% CP)	23.00	Crude protein	19.10
Decorticated cottonseed meal (45.6% CP)	8.00	Crude fat	4.02
Wheat bran	5.00	Calcium	1.00
Soybean oil	0.40	Phosphorus	0.60
Premix ^1^	5.00		
Total	100		

^1^ Premix was included per kg of diet: vitamin A, 250,000 IU; vitamin D, 60,000 IU; vitamin E, 1000 IU; Iron, 80 mg; Copper, 12 mg; Zinc, 80 mg; Manganese, 70 mg; Selenium, 0.40 mg; Iodine, 1 mg; Cobalt, 0.80 mg. The premix contains 4.2% calcium hydrogen phosphate. ^2^ Calculated value; NEmf (combined net energy) was a calculated value, which was the sum of the NEmf values for the ingredients multiplied by their percentages in the diets [19], while the other values were measured values.

**Table 2 animals-13-01570-t002:** Growth performance of calves receiving dietary supplementation with zinc amino acid complexes at different concentrations.

Parameters	Group A	Group B	Group C
Initial weight	69.23 ± 2.11	69.80 ± 1.88	69.00 ± 2.86
Final weight	118.92 ± 7.25 ^b^	125.58 ± 4.64 ^ab^	136.21 ± 9.12 ^a^
ADG/(kg/day)	0.82 ± 0.27 ^b^	0.93 ± 0.17 ^ab^	1.12 ± 0.18 ^a^
Daily dry matter intake/(kg/day)	0.88 ± 0.20	0.94 ± 0.29	1.03 ± 0.38
Feed/gain	1.07	1.01	0.92

ADG: average daily weight. Different lowercase letters within the same row indicate significant differences (*p* < 0.05).

**Table 3 animals-13-01570-t003:** Levels of serum immune indexes in calves receiving dietary supplementation with zinc amino acid complexes at different concentrations.

Indexes ^1^	Group A	Group B	Group C
IgA (μg/mL)	3407.42 ± 126.99	3396.66 ± 131.50	3273.79 ± 185.04
IgG (mg/mL)	6.21 ± 0.65	6.20 ± 0.73	5.27 ± 0.84
IgM (μg/mL)	1769.05 ± 134.82 ^a^	1862.19 ± 96.77 ^a^	1456.69 ± 114.49 ^b^
IL-2 (pg/mL)	827.99 ± 76.44 ^b^	913.02 ± 74.42 ^a^	746.37 ± 47.78 ^c^

^1^ Determined at three months of age. Different lowercase letters within the same row indicate significant differences (*p* < 0.05).

**Table 4 animals-13-01570-t004:** Levels of antioxidant capacity indexes in calves receiving dietary supplementation with zinc amino acid complexes at different concentrations.

Indexes ^1^	Group A	Group B	Group C
GSH-Px (U/mL)	135.56 ± 17.44	121.25 ± 10.64	125.22 ± 7.81
SOD (U/mL)	49.62 ± 8.33 ^ab^	58.02 ± 10.87 ^a^	42.68 ± 9.92 ^b^
T-AOC (μmol/mL)	0.93 ± 0.12 ^a^	0.94 ± 0.06 ^a^	0.74 ± 0.04 ^b^
MDA (nmol/mL)	6.08 ± 0.98 ^b^	6.59 ± 0.95 ^b^	8.33 ± 0.34 ^a^

^1^ Determined at three months of age. Different lowercase letters within the same row indicate significant differences (*p* < 0.05).

**Table 5 animals-13-01570-t005:** Data processing, statistical analysis, and quality control of sequencing results.

Sample ID	Raw Reads/Strip	Effective Reads/Strip	Q20 (%) ^1^	Q30 (%) ^2^	% Effective
A1	80,573	78,200	99.03	96.11	97.05
A2	79,834	77,586	99.03	96.1	97.18
A3	80,289	75,980	99.03	96.07	94.63
A4	80,222	78,239	99.04	96.12	97.53
A5	79,593	77,353	99.06	96.2	97.19
B1	79,923	75,746	99.06	96.18	94.77
B2	79,393	77,365	99.01	96.05	97.45
B3	79,864	76,888	99.05	96.15	96.27
B4	80,034	76,849	99.06	96.18	96.02
B5	80,122	78,441	99.07	96.2	97.9
C1	80,320	77,136	99.05	96.13	96.04
C2	80,238	78,192	99.1	96.33	97.45
C3	79,597	77,090	99.08	96.25	96.85
C4	80,105	77,420	99.08	96.24	96.65
C5	79,933	77,111	99.07	96.2	96.47
Total	1,200,040	1,159,596	nd	nd	nd

^1^ Q20 (%) refers to the percentage of bases with mass values greater than or equal to 20 based on the total number of bases; ^2^ Q30 (%) refers to the percentage of bases with mass values greater than or equal to 30 based on the total number of bases.

**Table 6 animals-13-01570-t006:** Alpha-diversity analysis of fecal samples of calves receiving zinc at different concentrations.

Parameters	Group A	Group B	Group C	SEM	*p*-Value
Richness indexes					
ACE	764.76 ± 56.42	782.44 ± 20.04	709.51 ± 66.53	57.65	0.11
Chao1	779.38 ± 61.14	803.52 ± 30.79	711.97 ± 66.44	64.87	0.06
Diversity indexes					
Simpson	0.98 ± 0.007 ^a^	0.98 ± 0.006 ^a^	0.95 ± 0.033 ^b^	0.23	0.02
Shannon	6.95 ± 0.39 ^a^	7.13 ± 0.29 ^a^	6.26 ± 0.49 ^b^	0.53	0.01

Different lowercase letters within the same row indicate significant differences (*p* < 0.05).

**Table 7 animals-13-01570-t007:** Relative abundances of gut microbiota at the phylum level in calves receiving zinc supplementation at different levels.

Microbial Phylum	Sample Groups			*p*-Value
	Group A	Group B	Group C	
*Firmicutes*	48.595 ± 4.340	50.912 ± 3.827	47.799 ± 6.854	0.63
*Bacteroidota*	42.671 ± 5.187	43.616 ± 4.286	45.222 ± 6.428	0.76
*Proteobacteria*	4.269 ± 2.791	2.384 ± 2.772	3.242 ± 1.572	0.50
*Spirochaetota*	3.197 ± 2.113	1.178 ± 1.258	1.597 ± 3.501	0.42
*Campylobacterota*	0.076 ± 0.072	0.137 ± 0.151	1.043 ± 2.190	0.43
*Verrucomicrobiota*	0.377 ± 0.482	0.627 ± 1.147	0.116 ± 0.155	0.55
*Patescibacteria*	0.215 ± 0.372	0.424 ± 0.172	0.283 ± 0.223	0.48
*Cyanobacteria*	0.158 ± 0.093	0.376 ± 0.293	0.350 ± 0.369	0.43
*Desulfobacterota*	0.288 ± 0.295	0.175 ± 0.142	0.298 ± 0.432	0.79
*Actinobacteriota*	0.082 ± 0.046	0.155 ± 0.140	0.038 ± 0.009	0.13
Others	0.071 ± 0.088	0.016 ± 0.013	0.010 ± 0.005	0.16

**Table 8 animals-13-01570-t008:** Relative abundances of gut microbiota at the genus level in calves receiving zinc supplementation at different levels.

Microbial Genus	Sample Groups			*p*-Value
	Group A	Group B	Group C	
*unclassified-Lachnospiraceae*	11.066 ± 0.919	8.940 ± 1.033	8.716 ± 2.320	0.07
*Alloprevotella*	6.229 ± 4.728	3.159 ± 4.212	9.431 ± 12.832	0.51
*Ruminococcaceae-UCG-005*	9.243 ± 3.686	11.167 ± 1.551	11.198 ± 4.050	0.57
*Rikenellaceae-RC9-gut-group*	5.694 ± 3.212	8.131 ± 4.052	5.364 ± 2.384	0.38
*Uncategorized-UCG-010*	2.870 ± 1.428	2.960 ± 1.368	1.648 ± 0.844	0.21
*Ruminococcus*	1.116 ± 0.339 ^a^	0.287 ± 0.142 ^b^	0.166 ± 0.134 ^b^	0.01
*Bacteroides*	4.081 ± 1.817 ^b^	9.694 ± 4.529 ^ab^	15.196 ± 5.423 ^a^	0.01
*Prevotellaceae-UCG-003*	3.836 ± 6.129 ^ab^	6.759 ± 5.184 ^a^	0.307 ± 0.426 ^b^	0.03
*Prevotella spp.*	1.115 ± 1.750	2.190 ± 2.159	2.096 ± 2.013	0.65
*Prevotellaceae-UCG-001*	0.066 ± 0.046 ^a^	0.103 ± 0.226 ^ab^	0.001 ± 0.001 ^b^	0.02
*Lactobacillus*	0.036 ± 0.022 ^a^	0.714 ± 0.531 ^b^	2.118 ± 2.956 ^ab^	0.01
*Colidextribacter*	0.049 ± 0.020 ^a^	0.292 ± 0.219 ^b^	0.301 ± 0.177 ^b^	0.05
*Faecalibaculum*	0.002 ± 0.001 ^b^	0.009 ± 0.004 ^a^	0.003 ± 0.004 ^b^	0.02
*Enterococcus*	0.002 ± 0.001 ^a^	0.018 ± 0.035 ^ab^	0.0006 ± 0.001 ^b^	0.04
*Bifidobacterium*	0.002 ± 0.003 ^ab^	0.013 ± 0.013 ^a^	0.001 ± 0.001 ^b^	0.03
Others	54.591 ± 1.661	45.564 ± 2.868	43.454 ± 7.849	0.68

Note: Different lowercase letters on the shoulder of peer data in the table indicate significant differences (*p* < 0.05), while the same lowercase letters on the shoulder of peer data in the table indicate no significant differences (*p* > 0.05).

## Data Availability

The datasets analyzed in the current study are available from the National Center for Biotechnology Information (NCBI) under accession number PRJNA783381. accessed on 13 December 2022 (https://www.ncbi.nlm.nih.gov/sra/PRJNA911430).

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
