# Peer review of "The Effect of Dietary Supplementation with Zinc Amino Acids on Immunity, Antioxidant Capacity, and Gut Microbiota Composition in Calves"

_animals, 2023, doi:10.3390/ani13091570_

Round 1

Reviewer 1 Report

Thanks to the authors for submitting this manuscript. The article is interesting but there are inconsistencies in the materials and methods that require a review.

These are my comments.

Line 85: The authors report 24 calves in the abstract divided into 3 groups and 30 calves in this line. The number is different.

Line 88: Animals were equally divided into three experimental groups 88 based on the level of Zn supplementation (eight animals per group). 24 or 30 calves?

Line 90: Please specify how many times Zn integration was done per day.

Line 93 : please correct punctuation.

Line 93: in the abstract authors write "The experiment ended when calves reached three months of age"(60 days) in this line 120 days. Please clarify this point as it is confusing.

Part 3.3:  "Antioxidant capacity". Please correct

Reviewer 2 Report

1- The manuscript focused on the effect of Zn -amino acid supplement at different concentrations and Zn content at the basal diet should know and analyzed.

2- There are a conflict regarding the calves number at this experiment because it is stated at the abstract 24 calves but in materials and methods it 30 calves????

3- You must present the details of the ingredients of basal diet e.g. The corn types and status,  the cotton seed meal corticated or uncorticated and bran

4- The equation of calculated net energy content must be presented

5- The premix percentage might be so high at the basal diet

6- The basal diet did not include sodium chloride or calcium carbonat although both of them are urgent at the diets of growing animals

Round 2

Reviewer 1 Report

Dear authors,

I appreciated the changes to the manuscript. In my opinion, the manuscript can be accepted for publication.

Author Response

Thank you for your suggestions. All your suggestions are very important. They have important guiding significance for my thesis writing and scientific research.

Thank you again for your advice.  Your suggestions have made the article more scientific and rigorous, which is of great significance to the manuscript. Thank you very much for your careful guidance.

Reviewer 2 Report

In general, no need to write significantly improved or enhanced or....and prefer to write improved (P<0,05)..

In lines 25-26, the supplementation dose should be 40,80 and 120 mg/kg DM

In line 132, need re-editing

In Table, cottonseed meal should be decorticated cotton seed meal

Does the CP content 46% cottonseed meal, Is it true??!!!

The premix does not contain sodium chloride?
